# Detection of Myocardial Infarction Using Hybrid Models of Convolutional Neural Network and Recurrent Neural Network

**Sumayyah Hasbullah [1,\*], Mohd Soperi Mohd Zahid [1,\*] and Satria Mandala [2]**

[1] Department of Computer and Information Sciences, Universiti Teknologi Petronas, Seri Iskandar 32610, Malaysia
[2] Human Centric Engineering & School of Computing, Telkom University, Bandung 40257, Indonesia; satriamandala@telkomuniversity.ac.id
\* Correspondence: sumayyah_19001019@utp.edu.my (S.H.); msoperi.mzahid@utp.edu.my (M.S.M.Z.)

**Abstract:** Myocardial Infarction (MI) is the death of the heart muscle caused by lack of oxygenated blood flow to the heart muscle. It has been the main cause of death worldwide. The fastest way to detect MI is by using an electrocardiogram (ECG) device, which generates graphs of heartbeats morphology over a certain period of time. Patients with MI need fast intervention as delay will lead to worsening heart conditions or failure. To improve MI diagnosis, much research has been carried out to come up with a fast and reliable system to aid automatic MI detection and prediction from ECG readings. Recurrent Neural Network (RNN) with memory has produced more accurate results in predicting time series problems. Convolutional neural networks have also shown good results in terms of solving prediction problems. However, CNN models do not have the capability of remembering temporal information. This research proposes hybrid models of CNN and RNN techniques to predict MI. Specifically, CNN-LSTM and CNN-BILSTM models have been developed. The PTB XL dataset is used to train the models. The models predict ECG input as representing MI symptoms, healthy heart conditions or other cardiovascular diseases. Deep learning models offer automatic feature extraction, and our models take advantage of automatic feature extraction. The other superior models used their own feature extraction algorithm. This research proposed a straightforward architecture that depends mostly on the capability of the deep learning model to learn the data. Performance evaluation of the models shows overall accuracy of 89% for CNN LSTM and 91% for the CNN BILSTM model.

**Keywords:** deep learning; convolutional neural network; long short-term memory; bidirectional long short-term memory; myocardial infarction

## 1. Introduction

According to the World Health Organization (WHO), cardiovascular disease has become the primary cause of death worldwide. From this, four of five cardiovascular disease death were due to myocardial infarction (MI) and stroke [1] MI or heart attack is a condition that occurs from decreasing oxygenated blood flow to a part of the heart resulting in the damage or death of the part. This mostly occurs due to coronary artery disease, which is also known as coronary heart disease. Unhealthy diet, physical inactivity, tobacco use, and the harmful use of alcohol are the main important risk factors that eventually lead to the disease [1] In the US, the American Heart Association state that every 40 s, an American will have a MI. Electrocardiogram (ECG) and cardiac enzyme tests can be used to detect MI. However, the cardiac enzyme can only be detected several hours after the attack and may show incorrect results if tested earlier. On the other hand, ECG provides faster results and helps in early intervention before further tests can be performed.

An ECG is a device that measures the electrical activity of the heartbeat. Cardiologists can detect abnormal conditions in some parts of the heart by measuring the amount of electrical activity passing through the heart muscle [2] The P wave is produced from the

sinoatrial node, which is the pacemaker of the heart and indicates atrial depolarization or atrial contraction [3]. The QRS wave represents the atrioventricular node and indicates ventricular depolarization or the contraction of the ventricle. The presence of the T wave shows ventricle relaxing or ventricular repolarization [3]. In the presence of MI, the ECG morphology would have a long ST interval, ST-segment elevation or depression and changes in the shapes of the T waves. The ST segment starts from the upward stroke called the J point after the S wave and ends at the start of the T wave. Figure 1 shows the ECG morphology of a normal or healthy heart.

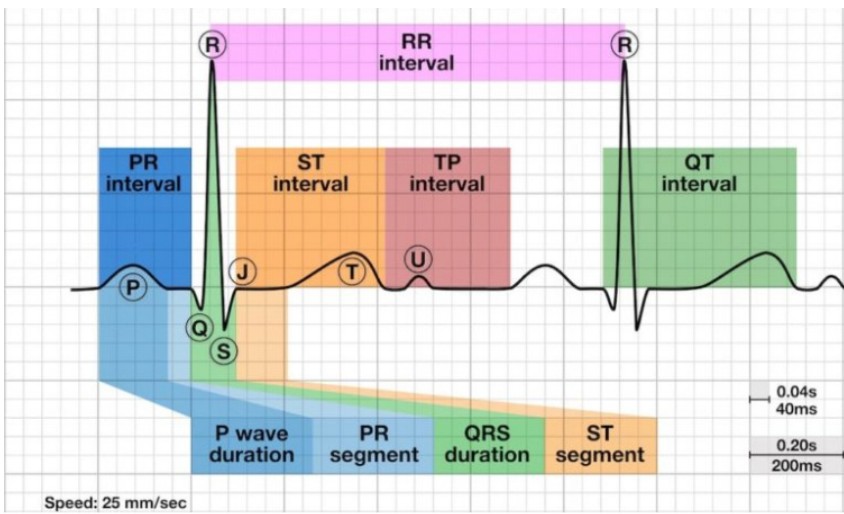

**Figure 1.** Shows the location of P, Q, R, S and T waves on ECG [4].

Manual analyses of ECG results by cardiologists are time-consuming and error-prone. In recent years, many attempts have been made to automatically detect MI from ECG signals. Various machine learning models have been used, such as K Nearest Neighbor [5,6], support vector machine [7–9], decision tree [6] and random forest [6]. Even though the results show good performance in predicting MI, the machine learning methods need handcrafted feature extraction, which requires high engineering and relies fully on expert knowledge to manually adjust the parameters [10].

There has also been notable research using deep learning to automatically detect MI. The use of deep learning models allows some pre-processing and feature extraction steps to be skipped as the model can learn the features by itself from the environment and its past mistakes [11]. This reduces the error and the time consumed when performed manually [12]. Several researchers have used a convolutional neural network (CNN) that automatically extracts spatial features [13–17] and long short-term memory (LSTM) that extracts temporal features [18,19]. In addition to that, there is also research that uses a hybrid of CNN and LSTM [20–22] to utilize the benefits of both features. There is also research [23,24] that combines CNN with Bidirectional LSTM, which is an improved version of LSTM that can capture the temporal features from the forward and backward of a time step.

Although previous research shows encouraging results, there is still room for improvement. Most research on MI detection produces outputs of binary classification. In real-life situations, patients may be diagnosed as having other cardiovascular diseases (CVD) instead of MI or free-from CVD. There is still less research that uses multiclass classification. The previous research with multiclass output includes additional processes to improve the performance, such as stacking decoding that uses more computational resources and manually provides feature extraction, which defeats the purpose of using deep learning to automate feature extraction. Thus, this research focuses on developing and comparing two hybrid models of CNN and RNN, which are CNN LSTM and CNN BILSTM models for automatic MI detection with multiclass classification output. No denoising technique and

signal modification were carried out on the raw ECG data. Therefore, the result depends on the deep learning models' capabilities to learn the pattern that differentiates the classes. In addition to CNN, which is well known for its ability to recognize spatial features, LSTM and BILSTM will help with processing the temporal features of the ECG data.

The contributions of this study are as follows:

- Identification of most relevant leads for MI detection from the PTB XL dataset for efficient use of computational resources in model development and training.
- Proposed models with straightforward architecture that better use the inherent capabilities of CNN and LSTM/BILSTM.
- Development of CNN LSTM and CNN BILSTM models to predict MI with three outputs: MI, Healthy and Other CVD.

To the best of our knowledge, this is the first study on MI detection with multiclass output using the PTB XL dataset rather than the PTB dataset, which is smaller in size. The rest of the manuscript is structured as follows: The related works are in Section 2. The material and method used are addressed in Section 3, followed by the results in Section 4. Section 5 contains the discussion, and Section 6 concludes this study.

## 2. Related Works

In this section, previous related works are discussed. The research [13] produces output classifications to 10 types of MI and a healthy control class using CNN utilizing 15 leads from the PTB dataset. The data were denoised using Discrete Wavelet Transform (DWT) and then fed to three CNN layers. A study by [14] also uses the PTB dataset with 12 leads and is classified into MI and healthy control classes. The ECG data were denoised using a moving wavelet filter, higher-order statistics, and morphological filtering to attend to different types of noises. After that, the data went through a feature extractor network, which consists of three layers of CNN for each lead. Then, all of the feature maps were combined and went through a feature aggregator network that consisted of one CNN layer for classification. Ref. [15] use only lead II of the PTB dataset as input to 11 layers of CNN and was classified into MI and healthy control. It also compared models with and without denoising techniques. In addition to that, ref. [16] uses 12 leads from the ECG-ViEW II dataset to classify MI and non-MI. The research used Synthetic Minority Oversampling Technique (SMOTE) to overcome data imbalance. The data were fed to a CNN and an RNN model for classification, with the CNN model showing better accuracy than the RNN model. In addition, to fit wearable ECG, a study by [17] used CNN to classify MI from healthy control using a single lead (lead I) from the PTB dataset. The research uses generative adversarial networks (GAN) to generate synthetic ECG data to overcome data imbalance and then fed the data into a CNN for binary classification of the MI and healthy control.

In [18], LSTM is used to classify anterior MI, inferior MI, and healthy control. This study used eight leads from the PTB dataset. All of the data went through denoising using a median filtering algorithm, a notch filter, and Chebyshev digital low-pass filter to remove different types of noises. Ref. [19] use LSTM to classify MI and healthy control using the PTB XL dataset lead II. The research first removed noise using a low pass filter and then went through four layers of LSTM.

Ref. [20] uses a hybrid model of CNN and LSTM, which produces a multiclass classification of MI for portable ECG devices using lead I from the PTB dataset and AF Challenge dataset. To remove noise, they used a Savitzky Golay filter. The study builds 35 layers of CNN and substitutes one of its fully connected layers with LSTM. It classifies the data into four classes, healthy control, MI, other CVD, and noisy class, for better classification that reflects the real-life situation. The accuracy result shows improvement after including stacking decoding, but unfortunately, stacking decoding increases the computational cost quadratically. Ref. [21] also used lead I of the PTB dataset with the aim of producing a portable ECG. The study consists of 16 layers of CNN, with LSTM as one of the layers. To overcome noise in the data, the study used the wavelet transform technique. It classifies

the data into healthy control and MI. Ref. [22] compares three deep learning techniques, CNN, LSTM and a hybrid of CNN and LSTM, using 12-lead ECG data from the Hospital of Sun Yat-sen University. In this study, CNN-LSTM ontained the best accuracy in classifying MI and healthy control as compared to the other two techniques. Ref. [23] developed a CNN-BILSTM using 12 leads of the PTB dataset and classified them as MI and the healthy control. BILSTM is formed by combining two LSTMs that capture the temporal features from forward and backwards at a specific time. In the research, they created a feature branch where each lead has a branch of feature extraction consisting of seven layers. Then, it goes through the lead random mask (LRM) for regularization to alleviate the effect of overfitting. Then, each branch will be fed into BILSTM, which has the same number of layers as the feature branch. Research by [24] use CNN BILSTM to classify a 12-lead PTB dataset into healthy, MI and non-MI classes. The data were first denoised using numerous filters. Then, the R peak and the other fiducial points were detected. The study extracted 21 features from the 12 leads of the ECG to help the system differentiate between the features of the different classes. This will also help to reduce the bias present in an imbalanced dataset. Then, the data were fed into a model with a layer of a CNN and a layer of BILSTM.

In summary, several deep learning models have been developed to detect MI. However, further studies should be undertaken to investigate the efficiency when some preprocessing steps are skipped, automatic feature extraction is utilized, different ECG leads are used, and a larger dataset is used, such as the PTB XL dataset.

## 3. Materials and Methods

In this section, the sequencing process of the model is discussed. The flowchart is as shown in Figure 2.

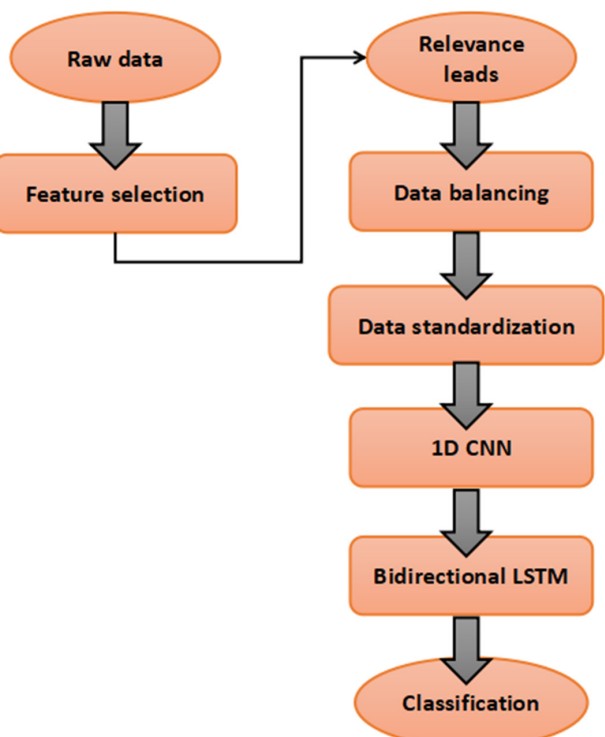

**Figure 2.** Flowchart of the proposed model.

*3.1. Dataset*

Several datasets are publicly available for use in terms of MI detection research, such as the ECG-ViEW II dataset, the PTB Diagnostic ECG dataset, and the PTB-XL dataset.

The dataset used in this research is the PTB XL dataset from the PhysioNet archive. The dataset is a large public dataset that contains 21,801 clinical 12-lead ECGs from

18,869 patients of 10-s length. The genders covered in this record are balanced, with 52% male and 48% female, and the age range is from 0 to 95 years. The dataset is reliable as all of the records were validated by a technical expert focusing on the signal characteristics and a cardiologist, with a large fraction of the records also validated by a second cardiologist. It is available for the public at a 16-bit precision with sampling rates of 100 and 500 Hz, depending on users' convenience. PTB XL was recorded using devices obtained from Schiller AG in a long-term project at the Physikalisch Technische Bundesanstalt (PTB) between October 1989 and June 1996. There are five superclasses in the dataset: normal ECG, MI, ST/T change, conduction disturbance, and hypertrophy [25]. The PTB XL dataset can be accessed from the PhysioNet website.

In this research, three selected leads of the ECG from the PTB XL of 10-s length were used. The leads are selected based on their relevance to MI detection. The chosen sampling rate is 100 Hz to fit the limited resources provided by the free version of the Google Colab Notebook. In addition to that, there are three classes used in this research, which are healthy (normal superclass), MI (MI superclass) and other CVD (combines the remaining 3 superclasses).

### 3.2. Data Cleaning and Selection of ECG Lead

Most research concerning MI detection uses 12-lead ECGs. According to [24], using 12-lead ECGs provides a better understanding of the inter-lead relationship and more information concerning the different parts of the heart, generating more features for training. However, there is research that used only certain leads and yielded good results. In this research, we explore using fewer leads but that are still relevant to MI detection. This is useful for modern ECG devices that are mobile, smaller in size, and have a wireless connection, which do not incorporate all 12 leads.

To demonstrate the leads that are most relevant to detecting MI from the PTB XL dataset, we ran the dataset through feature importance in random forest. Before the feature importance process, the data that contain 2 or more superclasses and the data with no superclass provided, were deleted from the dataset. After the cleaning process, the dataset was filtered through the feature importance, and three leads were chosen based on the results.

### 3.3. Data Balancing

After performing the data cleaning and lead selection processes, the amount of MI data is in the minority, as shown in Figure 3.

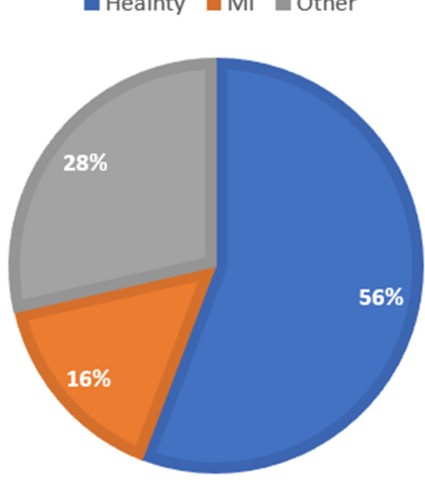

**Figure 3.** Distribution of classes.

Data balancing is a technique used to reduce bias. There are several ways to perform data balancing, as stated by [26], such as oversampling, undersampling and a hybrid of both over and undersampling. In this study, SMOTEEN hybrid data sampling was used. The minority data were first oversampled using SMOTE, and then the Edited Nearest Neighbor (ENN) was applied to undersample the majority data by removing the overlapping data. The overlapping data are chosen based on three of their nearest neighbors. If the data are misclassified by the three nearest neighbors, the data will be deleted.

### 3.4. Proposed Model

The proposed model was built by combining two neural networks. The first model combined CNN and LSTM, and the second model combined CNN and BILSTM to compare and achieve the benefits of both algorithms. The models were created layer by layer, starting from the CNN layer. After tuning the hyperparameters of the layer, the next CNN layer was created, and the hyperparameters were tuned. Different numbers of kernels ranging from 2 to 10 and different numbers of nodes with values of 16, 32, 64, 128, and 256 were experimented with to obtain the optimum values for the layer. After that, the LSTM/BILSTM layer was created by also tuning the hyperparameter using different nodes. To improve the performance of the model, we also experimented with the max-pooling layer, batch normalization layer, and dense layer. To maintain regularization, dropout layers experimented with values of 0.0 to 0.5 at different parts of the model. We also try to include L2 regularization. The process repeats until the optimum number of layers with the optimum hyperparameter is reached. We tried several configurations of CNN LSTM and CNN BILSTM before coming out with the final layer.

### 3.4.1. CNN LSTM

CNN is a feed-forward network and shows excellent results in terms of extracting spatial features and can thus be utilized to extract the short-term fluctuation features of ECG signals. The major difference between a CNN compared to a traditional artificial neural network (ANN) is that only the last layer of a CNN is fully connected for classification, but in an ANN, each neuron is connected to every other neuron [27]. This increases the tendency of an ANN to overfit when processing images due to the image size. Using a CNN, the raw pixel intensity of the input image transforms into a flattened vector before applying mathematical operation using an activation function on it in the hidden layers and predicts the class or label probabilities during the output. The neurons will only connect to a small region of neurons in the previous layer. In 1D CNN, the kernel slides along one dimension and suits the ECG signal dimension. Through the layers of a CNN, the filters are applied, and thus features from the basic to more complex of the ECG are learned. The equation to obtain the convolution output, $y$, is as follows [28]:

$$y_m = \sum_{k=-p}^{p} x_{m-k} w_k \tag{1}$$

where $x$ is the input signal, $w$ is the kernel, $p$ is the length of the kernel, and $m$ is the length of the signal.

The model CNN LSTM contains 2 layers of 1D CNN with 32 and 64 nodes each. Both used kernels that were fixed to 10. The ReLU activation function is applied, and each 1D CNN layer is followed by a batch normalization and a 1D Maxpooling layer to reduce overfitting. The output from the CNN is then fed into an LSTM layer as input with 8 nodes.

Earlier, the recurrent neural network (RNN) was developed to specialize in extracting the temporal features. In addition to learning forward, RNN also remembers things learned from the prior output and allows the information to persist. However, RNN is exposed to vanishing gradient problems when used in cases of long-term data. It is when the gradient shrinks as it back-propagates through time and eventually stops learning. As for that, long short-term memory (LSTM) is created, which is a version of RNN that is capable of learning long-term dependencies. LSTM consists of gates, as shown in Figure 4, which are capable

of learning which data in a sequence are needed or not, using the output of 0 and 1 from the sigmoid function.

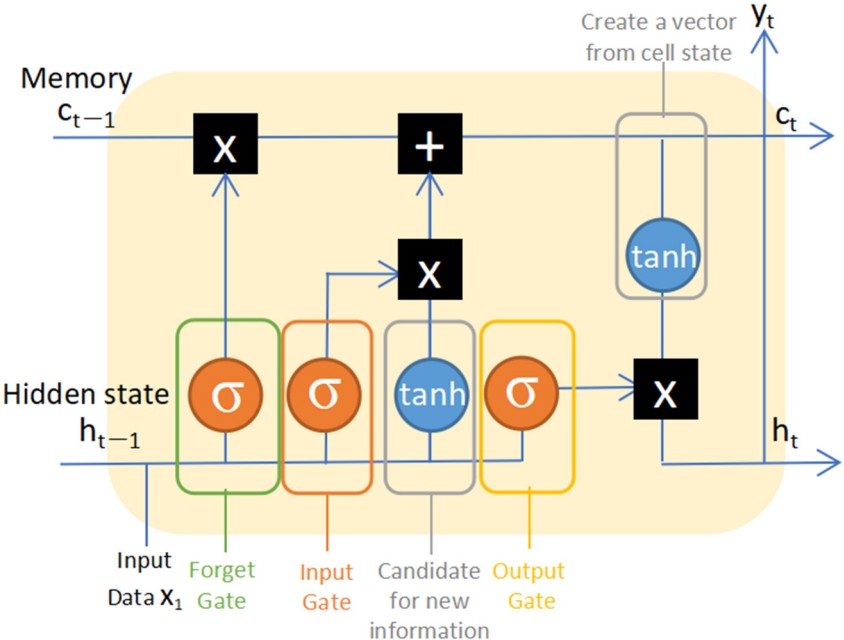

**Figure 4.** A long short-term memory (LSTM) cell state.

There are three types of gates in LSTM, the Forget gate, the Input gate, and the Output gate. Using that, relevant information will be passed through the long chain of the sequences called cell states to make a prediction. The inputs of the Forget gate are (1) the output (hidden state) from the previous cell and (2) the input from the current time step. Weight matrices are multiplied by the inputs, and a bias is added. After that, the sigmoid function is applied to the result. The output of the sigmoid function, which is between 0 and 1, will next be multiplied by the cell states. Through the forget gate, the output of 0 indicates that the piece of information is not needed and will be forgotten. In contrast, 1 indicates the information is needed and will be remembered. The Input gate is used to update the new information. The process is the same as the Forget gate, where a sigmoid function is applied to determine the new information that needs to be added. The tanh function creates a vector to all possible values that can be added with an output value from −1 to 1. Both outputs from sigmoid and tanh will be multiplied, and the result is added to the cell state by an additional operation. Next, the Output gate is responsible for selecting the best output from the cell gate, as not all information is suitable to be the output. The Output gate creates a vector after applying the tanh function to the cell state by scaling the value from −1 to 1. Using the sigmoid function, it creates a filter from the value of the hidden state and current input, and then, the result is multiplied with the output of the tanh function to regulate the values that need to be output from the vector. The output will be sent as the final output of the cell and to the hidden state of the next cell [29].

After going through LSTM, the data were classified through the dense layer with the Softmax activation function to three outputs which are: healthy, MI and other CVD. The summary of the CNN LSTM model is shown in Figure 5. The details of each layer with the parameters are shown in Table 1.

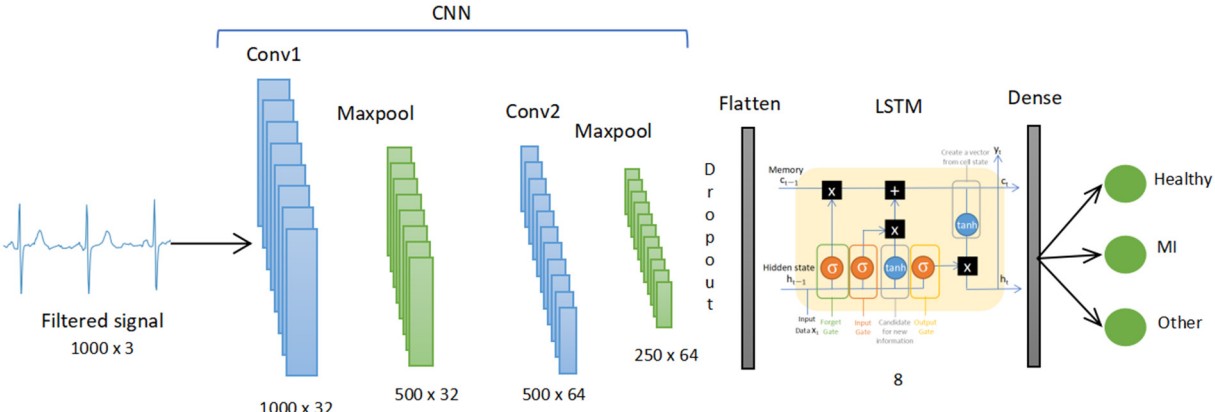

**Figure 5.** The CNN LSTM model.

**Table 1.** Details parameters of each layer in the CNN LSTM model.

| Layer Type | Output Shape | Parameter |
|---|---|---|
| Conv1d-1 | 1000,32 | 992 |
| Activation_1 | 1000,32 | 0 |
| Batch_normalization_1 | 1000,32 | 128 |
| Max_pooling1d_1 | 500,32 | 0 |
| Conv1d_2 | 500,64 | 20,544 |
| Activation_2 | 500,64 | 0 |
| Batch_normalization_2 | 500,32 | 256 |
| Max_pooling1d_2 | 250,64 | 0 |
| Dropout_1 | 250,64 | 0 |
| Time_distributed_1 | 250,64 | 0 |
| LSTM | 8 | 2336 |
| Dense_1 | 3 | 27 |

Total params: 24,283
Trainable params: 24,091
Non-trainable params: 192

### 3.4.2. CNN BILSTM

For the second model, the CNN was combined with the bidirectional LSTM (BILSTM). BILSTM functioned by connecting two LSTMs to capture the forward and backward information of a sequence at every time step, as shown in Figure 6. It is by adding one more layer of LSTM that catches information in the reverse flow. This helps capture future features instead of only the past features. By that, the input sequence is trained using two LSTMs where the first one with the normal LSTM and the other with the reverse copy of the input sequence [30]. Both final outputs will then be concatenated to form the final vector. This can provide further context to the network and result in quicker and deeper learning of the problem. It also helps improve feature extraction of the sequential time-series signal of an ECG.

The architecture of the CNN BILSTM model consists of 2 layers of CNN with 8 and 32 nodes each. The kernel was fixed at 10 for both. Each layer was followed by Maxpooling and batch normalization. Then, the data went through the BILSTM with 16 nodes. Then, Global Maxpooling was used to flatten the data to meet the size requirement of the dense layer. The dense layer with Softmax activation function then classified the data into 3 classes: healthy, MI, and the other CVD class. The summary of the CNN BILSTM model is shown in Figure 7. The details of each layer with the parameters are shown in Table 2.

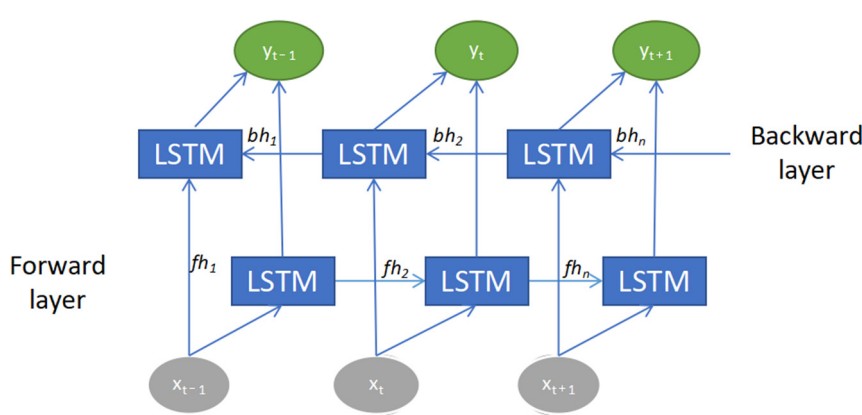

**Figure 6.** Bidirectional long short-term memory (BILSTM) architecture.

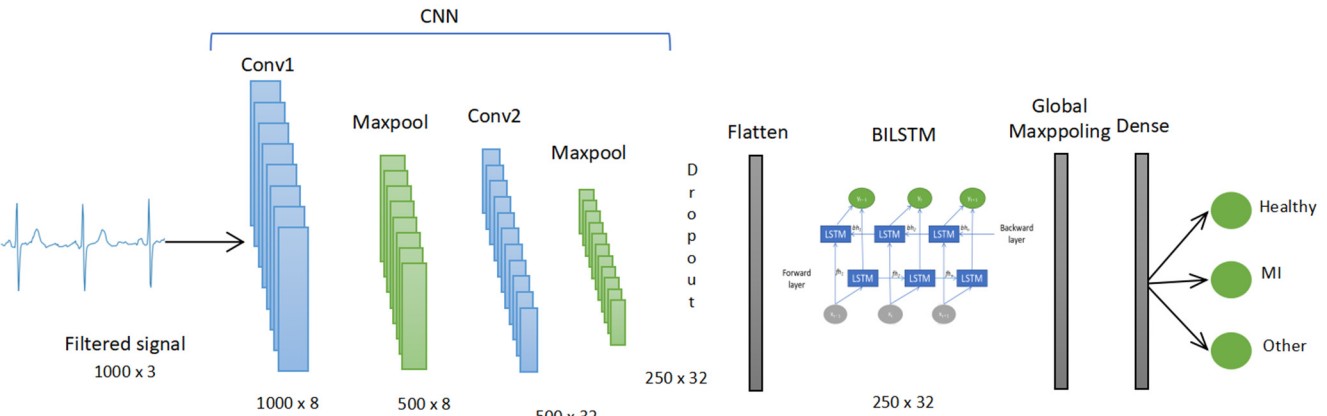

**Figure 7.** The CNN BILSTM model.

**Table 2.** Details parameters of each layer in the CNN BILSTM model.

| Layer Type | Output Shape | Parameter |
|---|---|---|
| Conv1d-1 | 1000,8 | 248 |
| Activation_1 | 1000,8 | 0 |
| Batch_normalization_1 | 1000,8 | 32 |
| Max_pooling1d_1 | 500,8 | 0 |
| Conv1d_2 | 500,32 | 2592 |
| Activation_2 | 500,32 | 0 |
| Batch_normalization_2 | 500,32 | 128 |
| Max_pooling1d_2 | 250,32 | 0 |
| Dropout_1 | 250,32 | 0 |
| Time_distributed_1 | 250,32 | 0 |
| Bidirectional_1 | 250,32 | 6272 |
| Global_max_pooling1d_1 | 32 | 0 |
| Dense_1 | 3 | 99 |

Total params: 9371
Trainable params: 9291
Non-trainable params: 80

### 3.5. Evaluation Metrics

The performance of the models was analyzed based on accuracy, precision, recall and F1-score.

The calculations of each matrix are as follows:

Accuracy: is the calculation of how many instances are accurately classified

$$Accuracy = (TP + TN)/(TP + FP + FN + TN) \qquad (2)$$

Precision: is the portion of relevant instances retrieved among the retrieved instances

$$\text{Precision} = TP/(TP + FP) \tag{3}$$

Recall: also known as sensitivity, is the measurement of how many relevant instances from the dataset are retrieved. In medical field-based research, sensitivity is an important matrix as it shows how much disease data the model is capable of detecting [31].

$$\text{Recall} = TP/(TP + FN) \tag{4}$$

F1-score: shows the capability of the model to accurately classify instances based on its class.

$$\text{F1-score} = 2 \times (\text{Precision} \times \text{Recall})/\text{Precision} + \text{Recall} \tag{5}$$

True positive (TP) means the model correctly predicts the positive class. True negative (TN) means the model correctly predicts the negative class. False positive (FP) shows that the model incorrectly predicts the positive class, and false negative (FN) shows that the model incorrectly predicts a negative class.

## 4. Results

There are two models proposed in this research, the CNN LSTM model and the CNN BILSTM model, which were developed to recognize the MI data from the healthy and the other CVD data. Both models were trained on 100 epochs with 32 batch sizes. For the loss function, categorical cross-entropy is used to match the multiclass output. Adam optimizer with a 0.001 learning rate is used for network weight. The data were divided into an 80:20 training-to-testing ratio, with a total of 10,483 pieces of training data and 2621 pieces of testing data. This section will cover the results of the training and testing.

### 4.1. Selection of ECG Leads

Concerning performing feature importance from a random forest, the results are shown in Figure 8. From the results, the three most relevant leads were chosen based on the highest position in the rank. The leads are V3, V4 and V6.

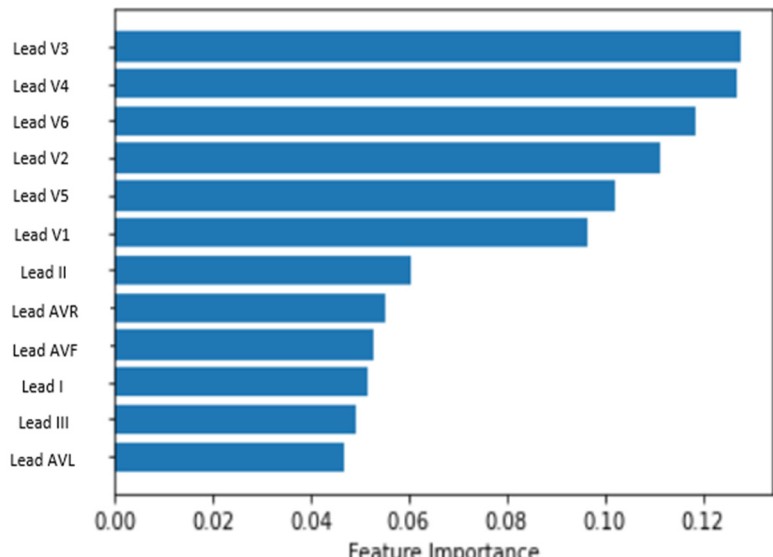

**Figure 8.** Result of the feature selection process.

### 4.2. Model Performance

The models were developed to differentiate MI from healthy and other CVD data using the CNN LSTM and CNN BILSTM models. Both models were developed using the

GPU of the Google Colaboratory (Colab) Jupyter Notebook. After training and validation, Table 3 shows the testing result of each performance parameter in classifying MI from other classes for both models. For CNN LSTM, the accuracy was shown to be 89%, and for CNN BILSTM, the accuracy was shown to be 91%.

**Table 3.** Result of each performance parameter.

| Performance Parameter | CNN LSTM Testing Result | CNN BILSTM Testing Result |
|---|---|---|
| Accuracy | 89% | 91% |
| Recall | Healthy: 48% MI: 97% Other: 75% | Healthy: 72% MI: 96% Other: 81% |
| Precision | Healthy: 52% MI: 89% Other: 92% | Healthy: 60% MI: 91% Other: 91% |
| F1-score | Healthy: 50% MI: 93% Other: 83% | Healthy: 65% MI: 93% Other: 86% |

The confusion matrix of both models is shown in Figure 9. It shows the percentage of instances that are correctly predicted compared to the incorrectly predicted instances. From the result, in both models, each class has a high percentage of correct predictions, with MI achieving the highest number of correct predictions. The models were developed layer by layer with different hyperparameters compared until the optimum results were achieved.

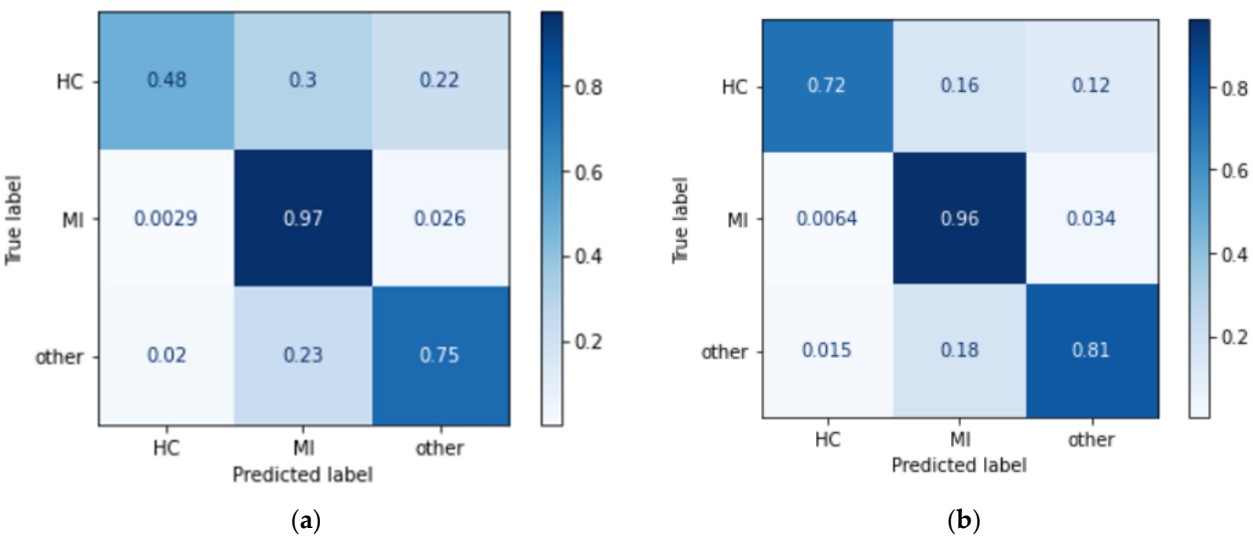

(**a**)          (**b**)

**Figure 9.** Confusion matrix of (**a**) CNN LSTM and (**b**) CNN BILSTM.

Figure 10 shows the accuracy and loss graph of the training and validation of the CNN LSTM model. Figure 11 shows the difference in terms of the training and validation of CNN BILSTM.

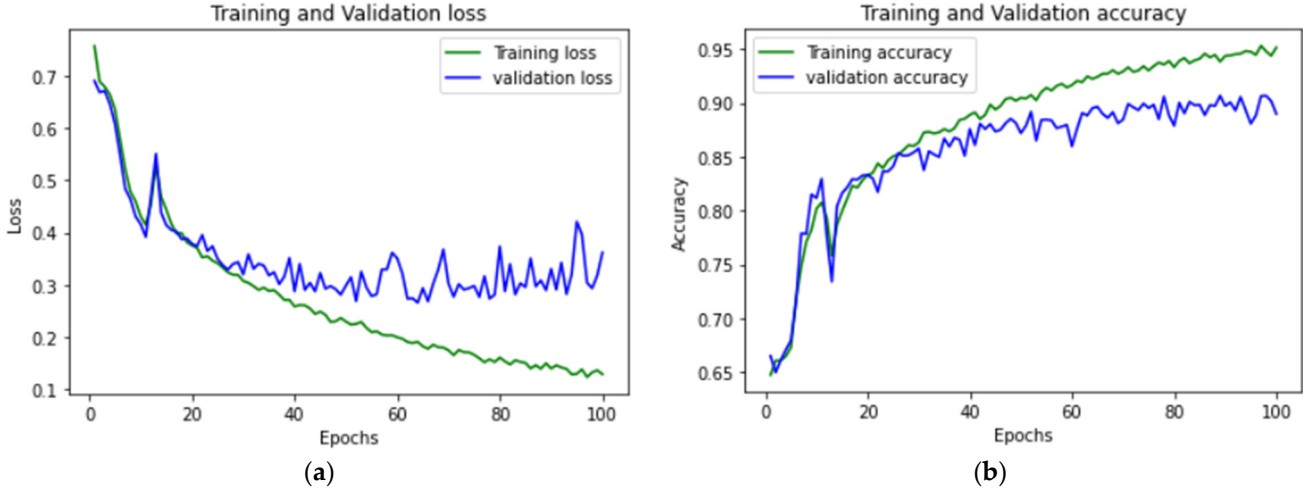

**Figure 10.** CNN LSTM (**a**) accuracy graph and (**b**) loss graph of training and validation.

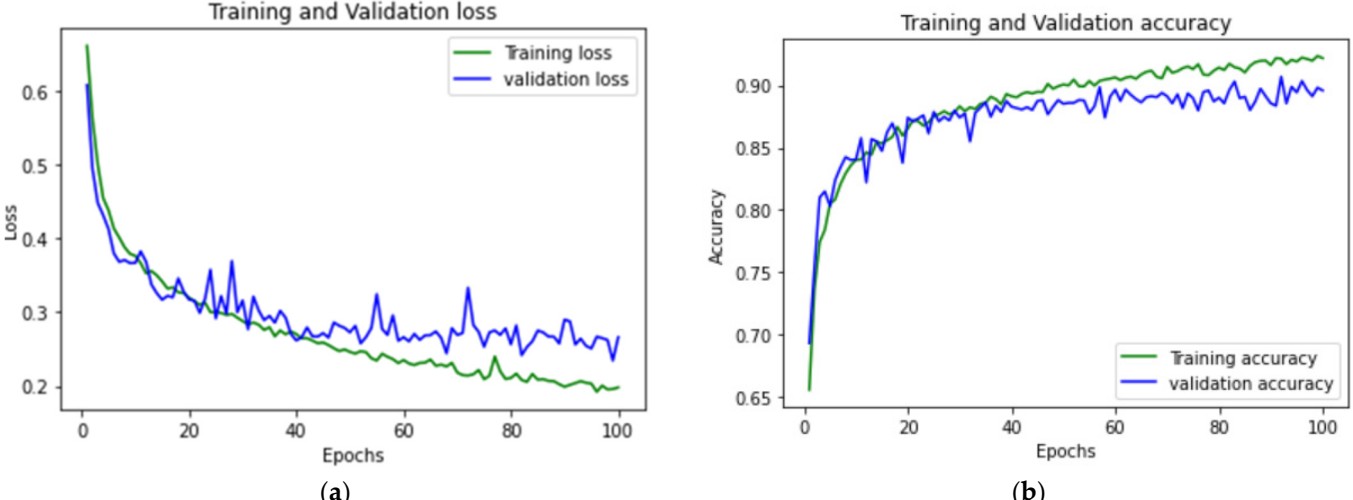

**Figure 11.** CNN BILSTM (**a**) accuracy graph and (**b**) loss graph of training and validation.

## 5. Discussion

In this research, two models were developed focusing on MI classification using the three most relevant ECG leads. The ECGs selected are V3, V4 and V6. After training with both models, the testing result of CNN BILSTM shows better performance in terms of classifying the data into the three classes. Despite both models showing comparable performance in MI detection, the CNN LSTM shows a decreased ability to classify the other two classes, especially the healthy class. This shows that the ability to learn from forward and backward information of the ECG using the BILSTM provides more features for the CNN BILSTM model to learn and results in better performance.

In developing automatic MI detection systems when using an ECG, much noticeable research has been carried out and has yielded great results, but most of this research concerns binary classification. However, in a real medical situation, the patient with a complaint may have other types of CVD instead of MI. Table 4 shows a summary of studies that include a class for other CVD on top of the MI and healthy classes. A study by [20] used CNN-LSTM with stacking decoding to differentiate MI and achieved a 94.6% F1-score in terms of MI detection, but without stacking decoding, the result shows 73% in terms of F1-score. The study also included experiments using only CNN and CNN with stacking decoding, showing lower results than the models with an additional LSTM. Unfortunately, stacking decoding requires high computing power. In addition to that, ref. [24] achieved 99.52% in terms of F1-score in MI detection using temporal features that

were manually extracted from the 12 leads of an ECG to reduce redundancy and class imbalance as input to a shallow CNN-BILSTM model. The extracted features also helped to aid in the classification of different classes. While in this study, the CNN BILSTM model was able to obtain a 93% F1-score in MI detection and is comparable to the other prior study as this study uses a straightforward and simpler model with no denoising and R peak detection, which is the same as real-time ECG data state. Therefore, it is much easier to integrate the system into medical devices. The system was also capable of learning by itself without any manual feature extraction process. This resulted in faster training time. In addition to that, the high performance of the system is also affected by the selection of ECG leads. Instead of random selection, selecting the most relevant leads provides better features needed for the classification.

**Table 4.** Summary comparison of the proposed model with the other prior multiclass research.

| No | Study | Architecture | Dataset | Additional Process | ECG Leads Used | Denoising | F1-Score |
|---|---|---|---|---|---|---|---|
| 1 | Hin Wan Lui et al. [20] | CNN | PTB and AF Challenge | - | Lead I | Yes | 59.70% |
| 2 | Hin Wan Lui et al. [20] | CNN with stacking decoding | PTB and AF Challenge | Stacking decoding | Lead I | Yes | 75.90% |
| 3 | Hin Wan Lui et al. [20] | CNN-LSTM | PTB and AF Challenge | - | Lead I | Yes | 73.00% |
| 4 | Hin Wan Lui et al. [20] | CNN-LSTM with stacking decoding | PTB and AF Challenge | Stacking decoding | Lead I | Yes | 94.60% |
| 5 | Monisha Dey et al. [24] | CNN-BILSTM | PTB | Manual feature extraction | 12 leads | Yes | 99.52% |
| 6 | Proposed model | CNN-BILSTM | PTB XL | - | Lead V3, V4 V6 | No | 93% |

## 6. Conclusions

MI is a deadly disease that requires fast treatment. Even though there are experts that can diagnose it manually, it may be exposed to human error, and it is also time-consuming. To aid medical experts, we proposed a CNN-BILSTM that helps differentiate MI from the healthy class and the other CVD class. The data used were produced by selecting leads that are most relevant for the MI detection from the PTB XL. The result shows significant achievement, with an accuracy of 91% in terms of general classification in addition to a recall of 96% and an F1-score of 93% in terms of MI detection. This study maintains simplicity with no signal modification and denoising technique. For future studies, a higher sampling rate can be used to improve the performance. In addition to that, implementing it on a portable ECG and a cloud-based platform will help the development of portable MI detectors and can be implemented in the IoT network in the future. The black box model does not provide explanations to support prediction results. One possible future work would be to improve the research using an explainable machine learning model. Another possible future work is incorporating the functional data analysis technique [32–34] to add results interpretability features.

**Author Contributions:** Conceptualization, S.H. and M.S.M.Z.; methodology, S.H., M.S.M.Z. and S.M.; formal analysis, S.H.; writing—original draft preparation, S.H.; writing—review and editing, M.S.M.Z. and S.M. All authors have read and agreed to the published version of the manuscript.

**Funding:** This research has been supported by the Yayasan Universiti Teknologi PETRONAS Fundamental Research Grant (YUTP-FRG) under Grant No. 015LC0-244.

**Data Availability Statement:** This study uses public dataset available at PhysioNet website.

**Conflicts of Interest:** The authors declare no conflict of interest.

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
