# Peer review of "Detection of Myocardial Infarction Using Hybrid Models of Convolutional Neural Network and Recurrent Neural Network"

_biomedinformatics, doi:10.3390/biomedinformatics3020033_

Round 1

Reviewer 1 Report

This research proposes hybrid models of CNN and RNN techniques to predict MI. Specifically, CNN-LSTM and CNN-BILSTM models have been developed. The PTB XL dataset is used to train the models. The models predict ECG input as representing MI symptoms, healthy heart conditions or other cardiovascular diseases. Performance evaluation of the models shows overall accuracy of 89% for CNN LSTM and 91% for the CNN BILSTM model. 

The novelty of the article in relation to the previous studies must be highlighted. Also, the language of the article should be improved.

How the PTB XL dataset can be accessed?

The quality and presentation of figures should be improved.

All mathematical equations should be explained a bit more.

What are the testing and training data sizes?

Which method is best and why?

Why low sampling is considered for the analysis?

Can you compare your results with functional time series analysis? See, for example, https://www.mdpi.com/2227-7390/10/22/4279

Please remove all typos.

Reviewer 2 Report

Minor Comments:

Figure 3: I would like to see empirical values in the pie chart

Major Comments:

1)     Deep learning needs a lot of data. Do you think the sample size is optimum to train the model.

2)     There is an inherent lack of transparency in these “black-box” algorithms. Do you think this will prevent reproducibility of your work?

3)     Was there a sensitivity analysis performed to decided the network architecture and topology? If not, how would you justify the sizes and number of layers?

4)     Could you please describe the PTB-XL data. Is it updated database?

5)     The accuracy of your proposed model is 6% less than one model and 1% less than the other models from other studies (Table 4). Your model does not provide denoising. Why do you think your model will make a difference in predicting MI. I believe other models would just as easily predict MI.               

Reviewer 3 Report

1. auto hyphenation can be removed.

2. What is the training-testing ratio?

3. Why feature selection required here?

4. What is the advantage of hybrid models CNN-LSTM, CNN-BiLSTM?

Round 2

Reviewer 1 Report

This research proposes hybrid models of CNN and RNN techniques to predict MI. Specifically, CNN-LSTM and CNN-BILSTM models have been developed. The PTB XL dataset is used to train the models. The models predict ECG input as representing MI symptoms, healthy heart conditions or other cardiovascular diseases. Performance evaluation of the models shows overall accuracy of 89% for CNN LSTM and 91% for the CNN BILSTM model. The authors addressed all my comments and I recommend accepting the article.

A minor check is required.

Reviewer 2 Report

Authors have revised the manuscript considerably. The paper can be published.